# PhotoAgent: Exploratory Visual Aesthetic Planning with Large Vision Models

## Abstract

With the recent fast development of generative models, instruction-based image editing has shown great potential in generating high-quality images. However, the quality of editing highly depends on carefully designed instructions, placing the burden of task decomposition and sequencing entirely on the user. To achieve autonomous image editing, we present **PhotoAgent**, a system that reshapes the paradigm of autonomous image editing. PhotoAgent autonomously reasons about the necessary edits, generates a robust action plan, and executes adjustments through a closed-loop process, all without requiring detailed step-by-step prompt engineering from the user. Our approach integrates large language models for intentional reasoning and dynamic planning, alongside a vision-language model for precise localized editing. This combination allows PhotoAgent to interpret users' aesthetic intents, decompose them into executable sub-tasks, and refine the output iteratively based on visual feedback. Extensive experiments demonstrate that PhotoAgent significantly outperforms existing methods in both instruction faithfulness and visual quality across a diverse range of editing scenarios.

## 1 Introduction

Recent instruction-based image editing models (InstructPix2Pix (Brooks et al., 2023a), SDXL (Podell et al., 2023), SD (Rombach et al., 2022), GPT-4o (OpenAI, 2024a), Flux.1 kontext (Labs et al., 2025), Bagel (Deng et al., 2025) etc.) enable amateur users to achieve professional photo edits through natural language commands (e.g., remove the passerby), rather than manipulating low-level sliders (e.g., brightness and color). This shift effectively redefines computational photography, extending its scope from ensuring *fidelity to the captured scene* to facilitating *fidelity to the user's aesthetic intent*, thereby democratizing powerful photographic expression (Wang et al., 2024a) (Liu et al., 2025).

Despite these advances, a critical bottleneck remains: these powerful models operate under a user-in-the-loop paradigm. Their effectiveness is largely contingent upon the user's ability to formulate precise, sequential instructions, which is difficult for end-users. As shown in Fig. 1, this reliance introduces several fundamental limitations: (1) *Expertise barrier*: Effective interaction requires expert knowledge. Amateur users often struggle either with designing articulate and precise editing instructions (e.g., decomposing "make my photo better" into low-level steps) or with evaluating whether editing results meet professional quality standards. (2) *Algorithm selection*: Different editing tasks require different specialized models. A single model may not be sufficient for all tasks, so users need to switch between models to achieve the desired results. (3) *Interaction complexity*: These models often require users, even professional ones, to issue multiple iterative commands, which is inherently laborious and prevents full automation for batch processing.

We argue that the next frontier in computational photography is not merely a powerful editor or processor (Brooks et al., 2023a) (Hertz et al., 2022) (Wang et al., 2024a) (Liu et al., 2025), but an autonomous editing agent that can enhance photos without requiring expert-level operation. Such an agent would emulate the decision-making process of a human photo editor, who strategically selects and sequences tools based on an assessment of the image's needs, and edits with specific tools. Recently, large vision and multimodal models (LVMs) (Deng et al., 2025) (Liu et al., 2023) (Brooks et al., 2023b) (Wang et al., 2024a) have demonstrated remarkable perception and instruction-conditioned editing capabilities, making an autonomous editing agent feasible. Our goal

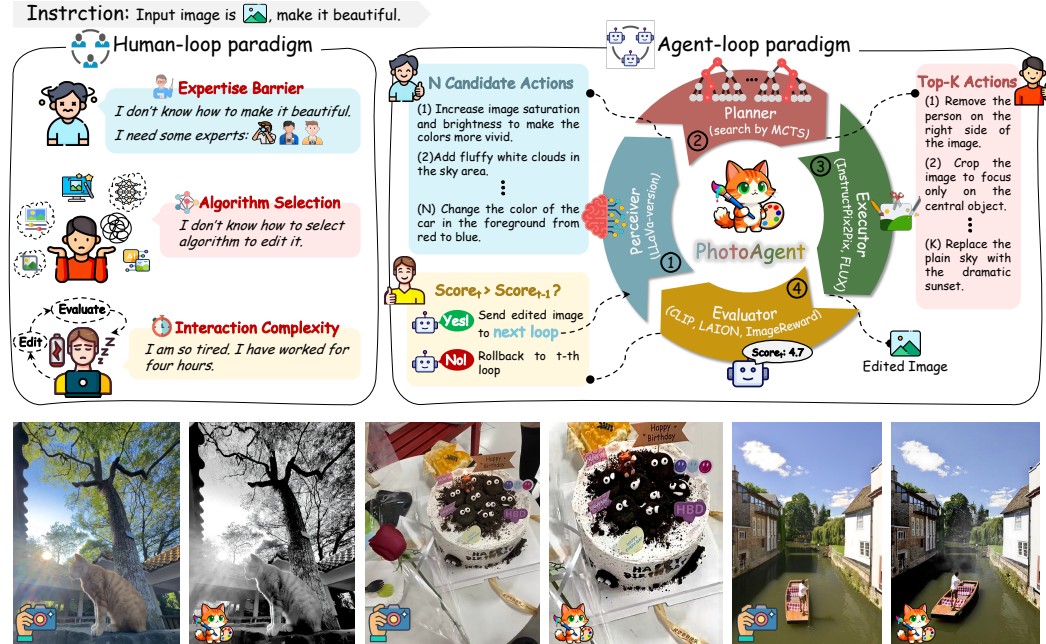

Figure 1: Overview of the PhotoAgent system. PhotoAgent performs high-level, semantically meaningful edits, moving beyond low-level color, contrast, or illumination tweaks. **Upper-Left**: Human-loop, where users iteratively inspect the image, propose edits, and apply changes until satisfied. **Upper-Right**: Agent-loop (ours), where the process runs autonomously: (1) the Perceiver proposes candidate actions, (2) the Planner (MCTS) searches for the best action, (3) the best action is executed by the Tools, and (4) the evaluator provides feedback for the next iteration. **Bottom**: Example photos before and after editing.

is to leverage these powerful tools to construct a coordinated, system-level framework, rather than innovating on the underlying editing models themselves.

In this paper, we introduce **PhotoAgent**, a novel autonomous system that integrates large vision and multimodal models (LVMs) with a suite of editing tools into a coherent framework. As illustrated in Fig. 1, PhotoAgent enables *more general and semantically meaningful editing*, moving beyond the low-level adjustments (e.g., color, contrast, illumination) that existing photo-editing agents such as JarvisArt (Lin et al., 2025), MonetGPT (Dutt et al., 2025), and 4KAgent (Zuo et al., 2025) primarily perform. This is achieved through programmatic control over flexible APIs and open-source editing platforms, allowing actions such as adding a sun to a dim sky or modifying objects within the scene. In addition, PhotoAgent introduces *exploratory visual aesthetic planning* within a *closed-loop* framework. Unlike open-loop systems (e.g., GenArtist (Wang et al., 2024b)) that execute linear action sequences without feedback, PhotoAgent continuously evaluates its edits and strategically explores the editing space. This helps to avoid both short-sighted decisions and irrecoverable artifacts that commonly happen in greedy approaches, enabling semantically meaningful, coherent, and high-quality enhancements.

Specifically, PhotoAgent consists of four core components: a **perceiver**, a **planner**, an **executor**, and an **evaluator**. The process begins with a VLM-based perceiver (e.g., LLaVA (Liu et al., 2023)) that interprets the input image and produces a set of semantically meaningful editing actions. These candidate actions are then passed to a Monte Carlo Tree Search (MCTS)-based planner (Chaslot et al., 2008) (Browne et al., 2012), which explores possible editing trajectories in a tree structure and selects the top-K most promising actions. This exploratory mechanism ensures that our system embodies **exploratory visual aesthetic planning**, avoiding myopic decisions. The selected actions are subsequently executed using either advanced image generation tools (e.g., Flux.1 Kontext) or traditional image processing libraries (e.g., OpenCV). Finally, an evaluator assesses the impact of each executed action, retaining only those that improve the current image state. By iterating through this

**perceive–plan–execute–evaluate** cycle, PhotoAgent forms a fully closed-loop process, enabling autonomous and reliable progress toward the final editing goal.

One major challenge is that existing image quality evaluation methods are not accurate enough for user-driven photo editing (also referred to as user-generated content, UGC). The core issue lies in the composition of existing datasets, where existing datasets are overly generic, containing a large proportion of AI-generated images, screenshots, advertisements, and posters, rather than authentic user-captured photographs. This discrepancy makes them insufficient for training models capable of guiding multi-step, user-preferred editing. To bridge this gap, we present the UGC-Edit dataset, a large-scale resource of multi-step editing tasks based on real user photos, generated via LLMs and rigorously verified by humans. We further train a reward model on this dataset to accurately assess visual quality, which plays a critical role in evaluating the system.

In summary, this work makes the following contributions:

- We propose PhotoAgent, an autonomous editing system that integrates a closed-loop architecture with a suite of editing and evaluation tools, enabling robust multi-step editing.

- We introduce a visual aesthetic planner to explore sequences of editing actions over long horizons, enabling deliberate, goal-driven image editing.

- We present the UGC-Edit dataset and introduce a reward model to support further research in autonomous image editing.

- Extensive experiments demonstrate that our complete system achieves significant improvements in editing quality.

## 2 RELATED WORK

**Image Editing** Early pioneering works primarily leverage Generative Adversarial Networks (GANs) (Goodfellow et al., 2014) or conditional encoder-decoder architectures for tasks like style transfer and attribute manipulation. For example, CycleGAN (Zhu et al., 2017) proposes unpaired image-to-image (I2I) translation, and StarGAN (Choi et al., 2018) enables multi-attribute manipulation within a single model. However, these approaches are inherently limited since their editing capabilities are confined to the narrow distribution of their training data, which often struggle with open-vocabulary requests. They frequently produce low-resolution or artifact-ridden outputs.

A paradigm shift was ushered in by the advent of powerful diffusion models (OpenAI, 2024b; Wu et al., 2025) and their integration with natural language. Models like Stable Diffusion (AI, 2024) treat image editing as conditional image generation, where the input image serves as a foundational condition. Recent methods (e.g., Prompt-to-Prompt (Hertz et al., 2022), InstructPix2Pix (Brooks et al., 2023a)) manipulate the features in latent space to enable highly flexible editing following open-vocabulary instructions. This progress continues with next-generation architectures based on flow matching (e.g., Flux (Labs et al., 2025)) and the integration of powerful Multimodal Large Language Models (MLLMs) like GPT-4o (OpenAI, 2024a), Show-o (Xie et al., 2024), Bagel (Deng et al., 2025), and Nano Banana (Google, 2025), which aim to tightly couple reasoning and generation. Despite these remarkable advances, a critical limitation exists. These models act primarily as single-step, static executors. Their performance is highly sensitive to meticulously engineered, low-level prompts, placing the burden of designing instructions and evaluations on the amateur user. These limitations prevent the method from handling complex, autonomous multi-step editing tasks, highlighting the need for a higher-level, planning-based framework.

**Planning with Autonomous Agents** To overcome the above limitations, a promising direction is to design an autonomous agent framework capable of multi-step planning and execution. Early works such as AlphaGo (Silver et al., 2016) employ planning algorithms like Monte Carlo Tree Search (MCTS) to navigate state spaces. Recently, LLM-based agents leverage LLM's reasoning capability to decompose tasks into sequences of actions (e.g., HuggingGPT (Shen et al., 2023), ReAct (Yao et al., 2023), and Voyager (Wang et al., 2023)).

Within computer vision, works have explored integrating planning into image editing tasks. Some approaches, such as JarvisArt (Liu et al., 2025) and MonetGPT (Dutt et al., 2025), leverage an LLM as a planner to parse a complex instruction into a sequence of calls to specialized image processing software. However, existing methods mainly focus on low-level editing tasks, such as color, tone, or

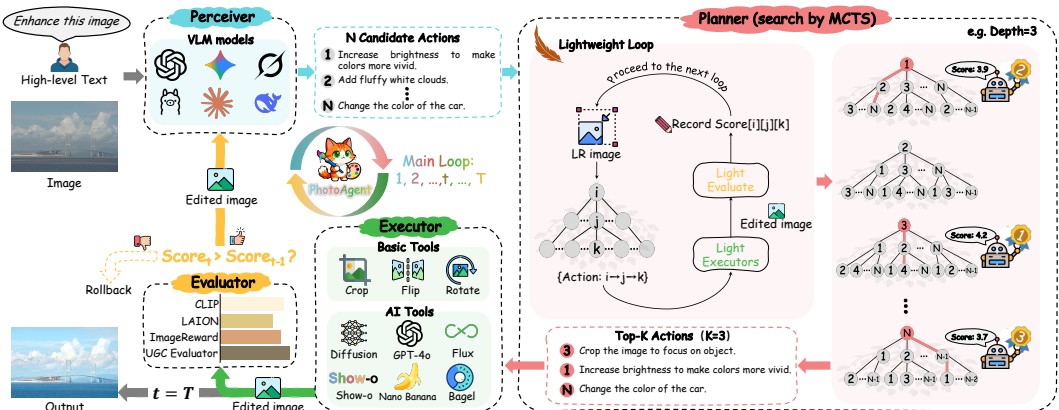

Figure 2: The proposed PhotoAgent, including Perceiver, Planner (MCTS), Executor, and Evaulator. First, Perceiver extracts semantic cues from the current image and proposes $N$ candidate editing actions. Second, Planner (MCTS) explores the candidate actions through iterative rollouts, scoring and pruning to progressively refine edits and select the sequence that achieves the optimal final result. Finally, the executor applies these edits while the evaluator scores intermediate results, invoking re-planning when the score is unsatisfactory.

exposure adjustments using procedural software tools like Lightroom or GIMP, which are limited to pure retouching.

More recent researches begin to explore directly applying MCTS and other search strategies to the text-to-image (T2I) generation process itself, building a search tree in the latent or textual space to find sequences of actions that better satisfy a high-level goal (Shi et al., 2025). However, existing methods (Lin et al., 2025; Zuo et al., 2025; Dutt et al., 2025) offer limited planning capability for instruction-based image editing. Our approach addresses this through an MCTS planner that considers internal simulation with external execution. We also employ a learned reward model trained on user preferences to guide the search. This combination enables robust planning with a diverse toolset and is supported by a new editing-specific benchmark for evaluation.

**Evaluation in Image**    In an automated image editing pipeline, the evaluator is important, as it defines the reward function that guides the agent's actions and determines the final output. Traditional full-reference image quality metrics, such as Peak Signal-to-Noise Ratio (PSNR) and Structural Similarity Index (SSIM) (Wang et al., 2004), are not suited for this open-world setting. They require a ground-truth target image, which is obviously impossible for creative editing tasks.

The community then turns to no-reference metrics, including distribution-based measures like Fréchet Inception Distance (FID) (Heusel et al., 2017), aesthetic predictors (discus0434, 2024), or CLIP-based image-text alignment scores (Radford et al., 2021). While a step forward, these metrics are often too broad to provide reliable, fine-grained signals for specific editing tasks on user-generated content (UGC). They cannot capture the subtle quality differences that are crucial in specific image editing tasks, such as aesthetic-oriented editing. To address this limitation, we introduce a specialized UGC evaluation dataset and train a reward model on the dataset. The reward model is adopted from a pretrained vision-language model (VLM) that contains inherent knowledge. The dataset and model enable learning of aesthetic evaluation, providing precise feedback to guide the agent toward high-quality results.

## 3    PHOTOAGENT

We propose PhotoAgent, an autonomous image editing system capable of executing multi-step editing tasks through a structured, closed-loop framework. As shown in Fig. 1, the system comprises four core components: a perceiver that interprets the input image and generates candidate editing actions, an MCTS-based planner that explores and selects potential editing actions, a set of Executor Tools that apply the edits, and an Evaluator that assesses the editing results. The system operates iteratively through a perceive–plan–execute–evaluate cycle, where an abstract controller manages the

whole loop. This closed-loop process continues until the editing objective is met or a termination condition is satisfied.

**Perceiver: Instruction Candidates Generation**    The perceiver utilizes a VLM, e.g., LLaVA (Liu et al., 2023), Qwen-VL (Bai et al., 2025b;a), to analyze the visual input $I_t$ and generate a set of $K$ diverse and atomic editing actions $\{a_t^k\}_{k=1}^K$. Specifically, the VLM is guided by a structured system prompt that instructs it to act as an "image editing perceiver". It outputs $K$ distinct instructions targeting composition-level or object-level operations in a strict JSON format, such as object removal, color adjustment, cropping, or effect application. For example, representative candidate actions are "Adjust the color balance to enhance the blue of the sky and the green of the water", "Crop the image to remove the top left corner, focusing on the river scene".

**Planner: MCTS-Based Action Exploration**    The planner chooses the candidate actions through a MCTS-based planning process, as shown in Fig. 2. Specifically, unlike existing methods that edit without planning, our planner enables the agent to simulate sequences of future edits, evaluating their long-term consequences before execution. This approach avoids short-sighted decisions and irreversible mistakes. To achieve this, MCTS consists of four phases: selection, expansion, simulation, and backpropagation.

In the *selection* phase, the algorithm traverses the tree from the root node using a tree policy that balances between exploring less-visited actions and exploiting actions with high average rewards. When reaching a leaf node, the *expansion* phase adds new child nodes representing potential editing actions. For example, when evaluating an action like "adjust the color balance to enhance the blue of the sky and the green of the water", it creates a new node to represent the resulting image state.

In the *simulation* phase, we evaluate editing actions efficiently using a fast-approximation environment. Since MCTS relies on numerous simulations, each involving action execution, state estimation, and quality assessment, we apply reduced-resolution processing and restricted diffusion sampling to make the process fast. This allows rapid previewing of outcomes while preserving essential visual and semantic information for evaluation.

Finally, during backpropagation, we calculate the reward value and propagate it back through the tree. This updates the visit count and average reward of each visited node, helping the selection phase make better decisions. After a number of simulations, the algorithm selects the action with the highest average reward or the most visits from the root node for actual execution.

**Executor: Action Execution**    Then, the executors actually run the selected actions on the image. In practice, we select the top-K actions rather than only the action with the highest score, which ensures robustness and avoids simulation inaccuracies in the previous step. For each action, our system selects between traditional operators, e.g., color adjustment or cropping via OpenCV/PIL (Bradski, 2000)), and advanced generative models, e.g., FLUX.1 Kontext (Labs et al., 2025) or Step1X-Edit (Liu et al., 2025). We then employ our reward model to evaluate all results, retaining only the highest-scoring output as the next state $I_{t+1}$. This approach ensures that our final decisions are grounded in real outcomes rather than simulated estimates, significantly improving the reliability of our editing trajectory.

**Evaluator: Outcome Evaluation**    The evaluator assesses the set of edited images $\{I_t^k\}_{k=1}^K$ produced by executing the top-K actions, and outputs each an assessment score $\{r_t^k\}_{k=1}^K$. Our method employs an ensemble evaluation strategy that integrates traditional no-reference metrics (such as NIQE (Mittal et al., 2012b) and BRISQUE (Mittal et al., 2012a)), modern instruction-based assessment (such as CLIP-based aesthetic scoring (Radford et al., 2021; Schuhmann et al., 2022) and instruction-following evaluation (Liu et al., 2023)), and customizable perceptual models (see Section 4), to provide a comprehensive evaluation.

The highest candidate score is compared against the score of the input image $I_{t-1}$. If an improvement is observed, the corresponding image is selected as the next state. Otherwise, the system reverts to $I_{t-1}$. The process terminates when the maximum number of steps is reached or updates no longer change the result.

We further include a controller that manages the whole loop. It is an abstract module that tracks the current editing state, invokes MCTS for planning, executes the selected actions, updates the state

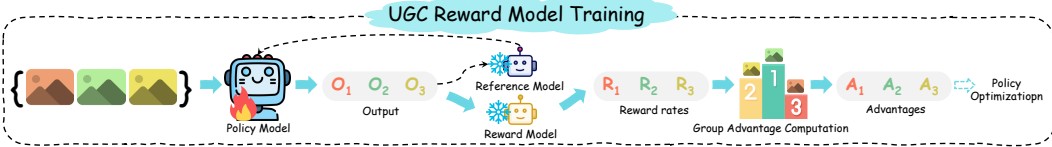

Figure 3: Pipeline for constructing the UGC-Edit Dataset. We start with a diverse pool of source images from LAION (Schuhmann et al., 2022) and RealQA (Li et al., 2025). Each image is processed through a structured prompt with Qwen (Wu et al., 2025) to generate diverse editing intents. The images are then filtered by human annotators. Finally, a reward model (Qwen2.5-VL 7B) is trained via GRPO (Shao et al., 2024) to predict fine-grained quality scores.

after each edit, and determines when to terminate the process (see Appendix A.5). This centralized rule-based design keeps the system modular and allows new tools or evaluators to be integrated without changing the core planning logic.

## 4  LEARNING A CUSTOMIZED EVALUATOR FOR EDITING SYSTEMS

Effective evaluation is critical for guiding autonomous image editing systems, where the output image should align with human aesthetic preferences. However, general metrics often fail to capture the aesthetic quality of user-generated photos, because they are designed for generic images and cannot fully capture aesthetic attributes. To address this gap, as shown in Fig. 3, we introduce a dedicated evaluation framework consisting of a custom preference dataset and a reward model to reliably assess photo quality in the editing system.

**UGC-Edit Dataset**  We construct the UGC-Edit dataset to provide high-quality supervision for UGC image evaluation. Images are sourced from the LAION aesthetic dataset (Schuhmann et al., 2022) and the RealQA benchmark (Li et al., 2025). Since LAION is a general web-image collection containing diverse image types, we apply a filtering process to select UGC images. Specifically, we first use a vision-language model (VLM) to tag image types and then conduct manual filtering. In this way, we obtain a subset of 7k images exhibiting authentic UGC characteristics. RealQA provides inherently UGC images with aesthetic ratings. We combine these two datasets and normalize the aesthetic scores to the same scale (1–5). For evaluation, we compile a separate test set of 89 images, including filtered samples and 20 photo groups captured ourselves to ensure ecological validity. This dataset provides the necessary bases for training a UGC assessment model aligned with user preferences.

**Reward Model Training**  We train a UGC reward model using the proposed dataset to predict fine-grained evaluation scores, which reflect human judgments of aesthetics on UGC photos. Specifically, we leverage Group Relative Policy Optimization (GRPO) (Shao et al., 2024) to optimize a pretrained VLM model (Qwen2.5-VL (Bai et al., 2025b)), as shown in Fig. 3. This strategy learns from relative rankings within groups of outcomes, improving robustness to annotation noise and variability. Consequently, the model captures subtle aesthetic preferences essential for photo editing and aligns with human aesthetic preferences. The resulting reward model provides reliable and high-quality evaluation signals that guide the entire photo-editing pipeline in the desired direction.

## 5  EXPERIMENTS

### 5.1  IMPLEMENTATION DETAILS

To ensure fair comparison under real-world conditions where users often provide ambiguous instructions, all methods receive the same vague editing prompt (e.g., "make this image better"). We choose

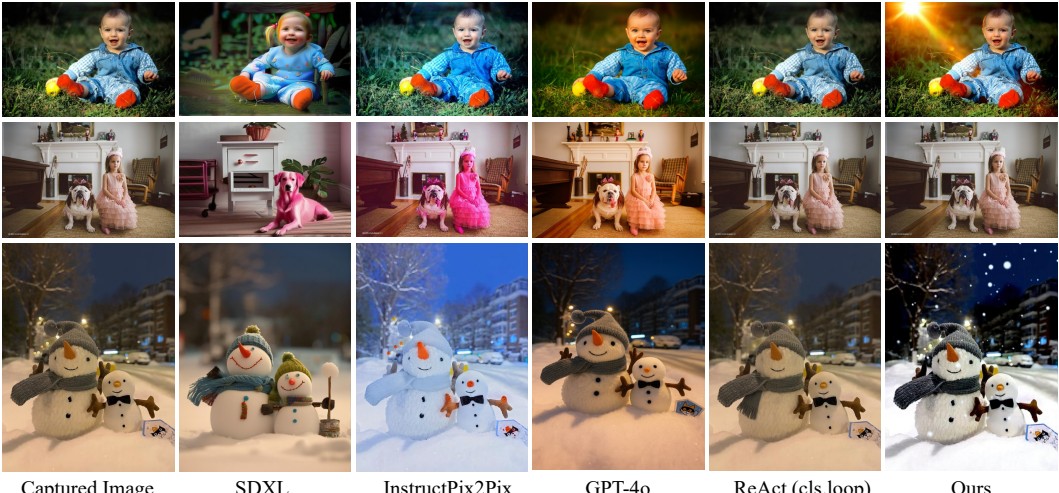

Captured Image      SDXL      InstructPix2Pix      GPT-4o      ReAct (cls.loop)      Ours

Figure 4: Qualitative results. Our PhotoAgent produces the most visually pleasing and goal-consistent edits. For example, it successfully corrects color, composition, and style in a coherent multi-step process, while baselines produce incomplete or flawed outputs.

Table 1: Quantitative comparison of different planning strategies on the UGC-Edit test set. The best results are in **bold**, and the second best are underlined.

| Method | CLIP Similarity (↑) | ImageReward (↓) | BRISQUE (↓) | Laion-Reward (↑) | UGC Score (↑) |
|---|---|---|---|---|---|
| Original Image | 0.6011 | 0.3836 | 0.6965 | 0.4876 | 3.022 |
| GPT-4o (OpenAI, 2024a) | **0.6142** | 0.4262 | 0.8763 | **0.5091** | **4.009** |
| InstructPix2Pix (Brooks et al., 2023a) | 0.6101 | 0.3984 | 0.6454 | 0.4751 | 3.207 |
| SDXL+Prompt (Podell et al., 2023) | 0.6084 | 0.3855 | 0.6955 | 0.4924 | 3.457 |
| Flux.1 Kontext (Labs et al., 2025) | 0.6059 | 0.4039 | 0.6718 | 0.4897 | 3.325 |
| HuggingGPT (Shen et al., 2023) | 0.6024 | 0.3844 | 0.7041 | 0.4841 | 0.367 |
| ReAct (Open-loop) (Yao et al., 2023) | 0.6024 | **0.3833** | 0.7041 | 0.4833 | 3.111 |
| ReAct (Closed-loop) (Yao et al., 2023) | 0.6026 | 0.3953 | 0.7030 | 0.4886 | 3.183 |
| PhotoAgent (Ours) | 0.6054 | 0.3954 | **0.6103** | 0.4934 | 3.465 |

two groups of baselines for comparison, including non-agent methods and agent methods. For non-agent methods, we compare with InstructPix2Pix (Brooks et al., 2023a), SDXL+Prompt (Podell et al., 2023), and Flux.1 Kontext (Labs et al., 2025), which performs editing in a single step without planning capabilities. For agent methods, we compare with HuggingGPT (Shen et al., 2023) (which generates all editing commands in a single call), ReAct (Yao et al., 2023) (Open-loop, iteratively plans and executes without evaluation), and ReAct (Yao et al., 2023) (Closed-loop, iteratively plans and incorporates an evaluator to decide action retention). This setup allows a comprehensive comparison across different architectural paradigms.

We calculate two types of metrics: semantic alignment and non-reference image quality. For semantic alignment, we use CLIP Similarity (↑) (Radford et al., 2021) to measure how well the edited image preserves the original content. For image quality, we report ImageReward (↓) (Xu et al., 2023) to approximate human preference alignment, BRISQUE (↓) (Mittal et al., 2012a) for non-reference image quality, and Laion-Reward (↑) (Schuhmann et al., 2022) for general aesthetic preference. Additionally, we report UGC Score (↑), which is calculated from our reward model fine-tuned on user-generated content (Section 4) to better reflect subjective user preferences.

## 5.2 MAIN RESULTS

**Quantitative Results** We show the quantitative results in Table 1. It can be seen that PhotoAgent achieves the best BRISQUE score, which reveals the effectiveness of our framework. In contrast, GPT-4o exhibits an over-editing intent, often outputting overly vivid colors and exaggerated contrast. While such outputs may score well on perceptual metrics, they can introduce significant image distortions. In addition, our method attains competitive performance on other metrics such as LAION-Reward, reflecting enhanced aesthetic appeal. As for agent-based baselines, their

Table 2: Correlation analysis between automated metrics and human preferences. Best results are in **bold**.

| Metric | PLCC ($\uparrow$) | SRCC ($\uparrow$) | KRCC ($\uparrow$) |
|---|---|---|---|
| ImageReward (Xu et al., 2023) | 0.4247 | 0.4198 | 0.3079 |
| Laion-aesthetic (Schuhmann et al., 2022) | 0.5567 | 0.5520 | 0.4068 |
| NIQE | 0.2787 | 0.3127 | 0.2202 |
| Qwen2.5-VL-7B (Wu et al., 2025) | 0.3239 | 0.3087 | 0.2735 |
| UGC evaluator (Ours) | **0.8300** | **0.8622** | **0.7525** |

overall performance is limited. In open-loop settings, this is mainly because they lack visual feedback, which can cause errors to accumulate and the system to drift away from the correct trajectory. In closed-loop settings, their performance is also constrained, which may lead to suboptimal or short-sighted decisions. The results demonstrate the effectiveness of the PhotoAgent, especially in producing consistent improvements in both semantic alignment and aesthetic quality.

**Qualitative Results** We also show qualitative comparisons in Fig. 4 to clearly demonstrate PhotoAgent's effectiveness. We observe that non-agent methods, such as GPT-4o, often apply generic edits and fail to address specific issues when given vague instructions (e.g., "make this image better"). Meanwhile, we find that agent-based baselines (Shen et al., 2023; Yao et al., 2023), often suffer from error accumulation and make short-sighted planning decisions, resulting in unsatisfactory visual output. In contrast, PhotoAgent effectively explores multiple editing paths through a closed-loop planning mechanism, and progressively selects and executes the most appropriate editing actions. As shown in Fig. 5, PhotoAgent first adjusts the overall tone to significantly enhance the image's aesthetic quality. Based on this, PhotoAgent can further edit specific objects, such as flying birds, which makes the scene appear more lively and dynamic. As a result, we achieve balanced outcomes that both preserve content and improve aesthetics.

### 5.3 EVALUATOR BENCHMARKING AND ANALYSIS

To validate the effectiveness of our customized evaluator, we compare its correlation with human judgments against existing metrics including ImageReward, NIQE, Laion-aesthetic, and Qwen2.5-VL. We report three correlation measures: PLCC (Pearson Linear Correlation Coefficient), SRCC (Spearman Rank Correlation Coefficient), and KRCC (Kendall Rank Correlation Coefficient). Table 2 shows that our UGC evaluator achieves superior alignment after GRPO fine-tuning, significantly outperforming all baseline metrics. This demonstrates its enhanced capability in capturing human aesthetic preferences, providing a reliable foundation for closed-loop evaluation in autonomous editing systems.

### 5.4 ABLATION STUDIES

We perform ablation studies to verify the key designs of PhotoAgent. We examine the effect of the UGC evaluator by removing it from the framework, which significantly decreases aesthetic metrics like Laion-Reward. This confirms the reward model's effectiveness in editing preferences. In addition, we investigate the importance of simulation times. Limiting the number of MCTS simulations to 10 results in suboptimal decisions, which demonstrates the necessity of strategic planning. Likewise, reducing the MCTS search depth to 1 can effectively implement greedy selection, leading to lower performance on multi-step edits. Overall, our results indicate that the evaluator, the depth of planning, and the number of simulations all play important roles in PhotoAgent's performance. Additional details are provided in Appendix A.1.

### 5.5 ANALYSIS

**Running Time** Although improving runtime is not the main focus of this work, we still report the inference time for reference. The running time of our system is influenced by multiple factors, including the number of simulations, the choice of executors, and GPU utilization. Under our standard configuration (search depth of 3, maximum of 20 simulations per iteration, and 3 editing iterations), the average processing time is approximately 470 seconds per image. Detailed parameter specifications and additional timing breakdowns are provided in Appendix A.1.

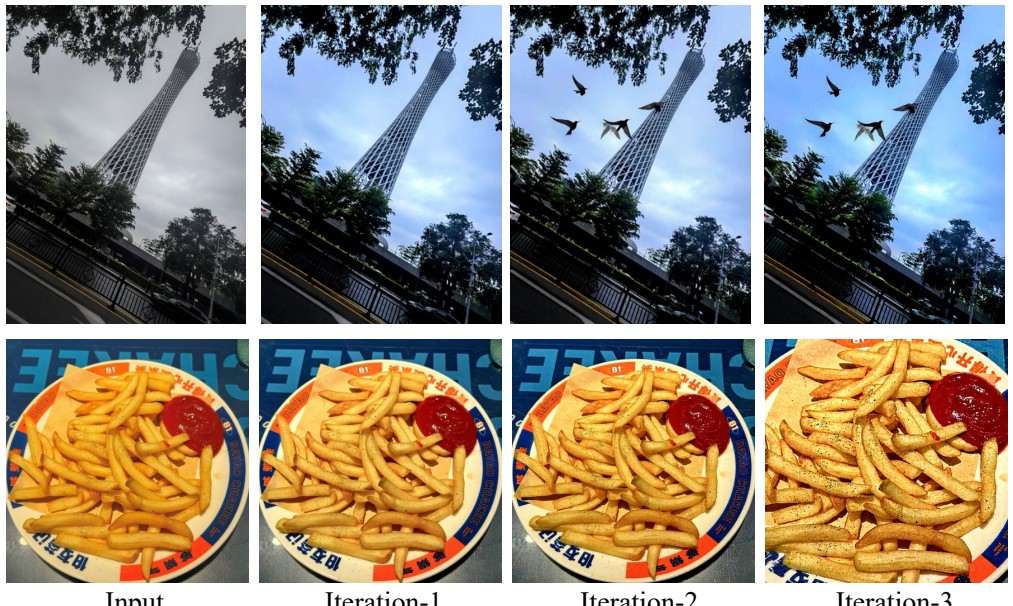

Input     Iteration-1     Iteration-2     Iteration-3

Figure 5: The editing process of our PhotoAgent over three iterations.

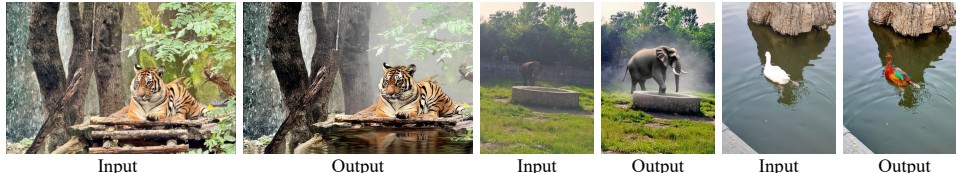

Input    Output    Input    Output    Input    Output

Figure 6: Some failed results where the editor may have made excessive changes.

**Failure cases** While our system performs well overall, it still exhibits several failure cases, as shown in Fig. 6. For example, dark or low-quality images can lead to unsatisfactory edits because the model struggles both to interpret the content and to apply effective adjustments. Moreover, when the input image is already of high quality, the system may introduce changes that add little value or, in some cases, refuse to make edits altogether. We also observe situations where the system makes technically reasonable modifications, but the codification cannot match user expectations, which is a general problem in image editing tasks. Please see more details in Appendix A.7 for potential solutions.

## 6 CONCLUSION AND FUTURE WORK

We present PhotoAgent, an autonomous image editing system that reframes photo editing as a sequential decision-making process, reducing the reliance on precise human instructions. The novelty arises from the coordinated interaction of multiple modules rather than innovating on the underlying editing models themselves. Specifically, the system integrates four key components: an LLM-based perceiver, an MCTS-driven exploration strategy, a tool-based executor, and a VLM-based evaluator, forming a closed-loop framework supported by the proposed UGC-Edit dataset. Experimental results demonstrate that PhotoAgent outperforms existing methods in producing semantically coherent and aesthetically consistent enhancements. The framework provides a foundation for more intent-aware editing in future work.

For future work, we will mainly focus on improving the system's overall user experience. First, the current system still has considerable room for improvement in speed, so adopting faster editing models and lighter search strategies could noticeably reduce inference time. Second, the system's robustness could be further enhanced by developing more personalized reward models that better capture diverse user preferences. Finally, we hope to broaden the scope of the system by expanding the action space with more detailed and creative editing tools, enabling the agent to tackle a wider range of tasks and application domains.

**Ethics Statement** This work adheres to ethical standards, using only publicly available data and ensuring no personally identifiable or harmful content is involved.

**Reproducibility Statement** We have made every effort to ensure that the results presented in this paper are reproducible. All code, weights, and the dataset will be released upon publication.

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

# A APPENDIX

## A.1 COMPUTATIONAL COST

To complement the discussion in Section 5.5, we provide a more detailed analysis of PhotoAgent's computational profile and the practical points that influence latency. Our goal is to make clear where most of the cost arises and provide a potential way to optimize in practice.

**Profiling the System.** We conduct a full profiling pass under the default configuration. As summarized in Table 3, the majority of the latency comes from the MCTS-based planner, where simulation and in-loop execution dominate the cost. Evaluator calls contribute a smaller but still noticeable fraction, whereas the perceiver stage contributes only a minor overhead. After removing the one-time model initialization (30s) and the duplicated evaluator calls counted inside MCTS, the total runtime is approximately 470s. For comparison, agent-based methods such as ReAct (cls.) have an inference time of approximately 120s, which is on the same order of magnitude.

Table 3: Runtime breakdown of PhotoAgent under the default configuration.

| Component | Time (s) | Percentage (total/parent) |
|---|---|---|
| Perceiver | ∼10 | 2.1% |
| Planner (MCTS) | ∼250 | 53.2% / 100% |
| $\rightarrow$ Executor (MCTS) | ∼170 | 36.2% / 68.0% |
| $\rightarrow$ Evaluator (MCTS) | ∼80 | 17.0% / 32.0% |
| Executor | ∼180 | 38.3% |
| Evaluator | ∼30 | 6.4% |
| Total[†] | ∼470 | 100% |

[†]Excludes initialization and duplicated evaluator time.

Here we provide three directions that may accelerate the system.

**MCTS Search Budget.** The first factor is the search budget of MCTS. Table 4 shows how varying the number of simulations directly trades off runtime and performance. Reducing the simulation count from 20 to 5 decreases the runtime from roughly 250s to 60s, while maintaining comparable BRISQUE, LAION-Reward, and UGC human scores. These results indicate that the default setting emphasizes quality, not speed, and that significantly faster operating points are readily attainable without architectural change.

Table 4: Impact of simulation budget on accuracy and runtime.

| Simulations | Time (s) | BRISQUE↓ | LAION Reward↑ | UGC Score↑ |
|---|---|---|---|---|
| 5 | ∼60 | 0.6723 | 0.4816 | 3.217 |
| 10 | ∼120 | 0.6522 | 0.4865 | 3.240 |
| 15 | ∼185 | 0.6429 | 0.4830 | 3.368 |
| 20 | ∼250 | 0.6103 | 0.4934 | 3.465 |

**Changing Editing Model.**  Second, an equally important factor is the choice of editing tool. Because PhotoAgent is tool-agnostic, its execution time can immediately benefit from faster generative models without structural modification. For example, at the same resolution (1080p), Step1x-Edit requires only about half the runtime of Flux.1 Kontext-Dev (reducing the time from ∼20s to ∼10s). Replacing the editing backend is a one-line API change, underscoring that the latency is not intrinsic to the framework.

**Model Acceleration.**  Finally, the system naturally benefits from standard model-optimization techniques used in production environments. Quantizing the transformer blocks of FLUX.1 Kontext to FP8 or FP4 yields over $2\times$ memory reduction and provides noticeably faster inference on NVIDIA Blackwell GPUs (NVIDIA Corporation). Comparable improvements can be obtained through TensorRT compilation or by adopting lower-precision evaluator models. These optimizations do not require any modifications to the PhotoAgent algorithm.

## A.2 SIM-TO-REAL GAP IN LOW-RESOLUTION MCTS SIMULATION

PhotoAgent uses reduced-resolution rollouts to make MCTS planning computationally feasible. This raises the question of whether an action sequence that scores well in low-resolution simulation remains optimal when executed at full resolution. To address this, the system incorporates several design choices that effectively control the sim-to-real gap.

**Evaluator Consistency Across Resolutions.**  The Evaluator exhibits stable scoring behavior between simulated and real environments. Table 5 reports the alignment between reward rankings computed at reduced resolutions and those obtained at full resolution. Even at one-quarter resolution, the top-ranked decisions are largely preserved, and rank correlations remain high.

Table 5: Consistency between simulated (low-resolution) rewards and full-resolution rewards.

| Metric | 1/2 resolution | 1/4 resolution |
|---|---|---|
| Top-1 retention (same best) | 85% | 75% |
| Top-3 retention | 100% | 90% |
| Spearman correlation | 0.94 | 0.79 |
| Kendall $\tau$ | 0.90 | 0.73 |

**Full-Resolution Re-Scoring of Top-$K$ Candidates.**  To further reduce sensitivity to coarse simulation, MCTS retains only the top-$K$ candidate actions and forwards them for full-resolution evaluation. The final action is selected using these high-fidelity scores. This step ensures that occasional deviations in low-resolution reward estimation do not influence the actual decision executed by the system.

**Closed-Loop Replanning After Each Executed Edit.**  The system applies only one action at a time. After executing the chosen edit at full resolution, MCTS restarts from the updated image. This closed-loop design prevents any discrepancies between simulation and execution from accumulating across steps, ensuring that each decision remains grounded in the real environment.

## A.3 GENERALIZATION EVALUATION ON EXTERNAL DATASET

To test how well our reward model generalizes beyond the UGC-Edit dataset, we evaluate it on the PARA dataset (Yang et al., 2022), which includes a wide variety of content, styles, and lighting conditions. Table 6 shows the correlation between the model's predictions and human aesthetic judgments. The model achieves SRCC scores around 0.75, surpassing prior state-of-the-art PIAA

models (Zhu et al., 2023), which attain roughly 0.70–0.72. These results demonstrate that the reward model consistently aligns with human preferences across other scenarios.

Table 6: Correlation of the reward model with human judgments on the PARA dataset.

| Metric | Aesthetic | Content |
|--------|-----------|---------|
| PLCC   | 0.7390    | 0.7577  |
| SRCC   | 0.7560    | 0.7702  |

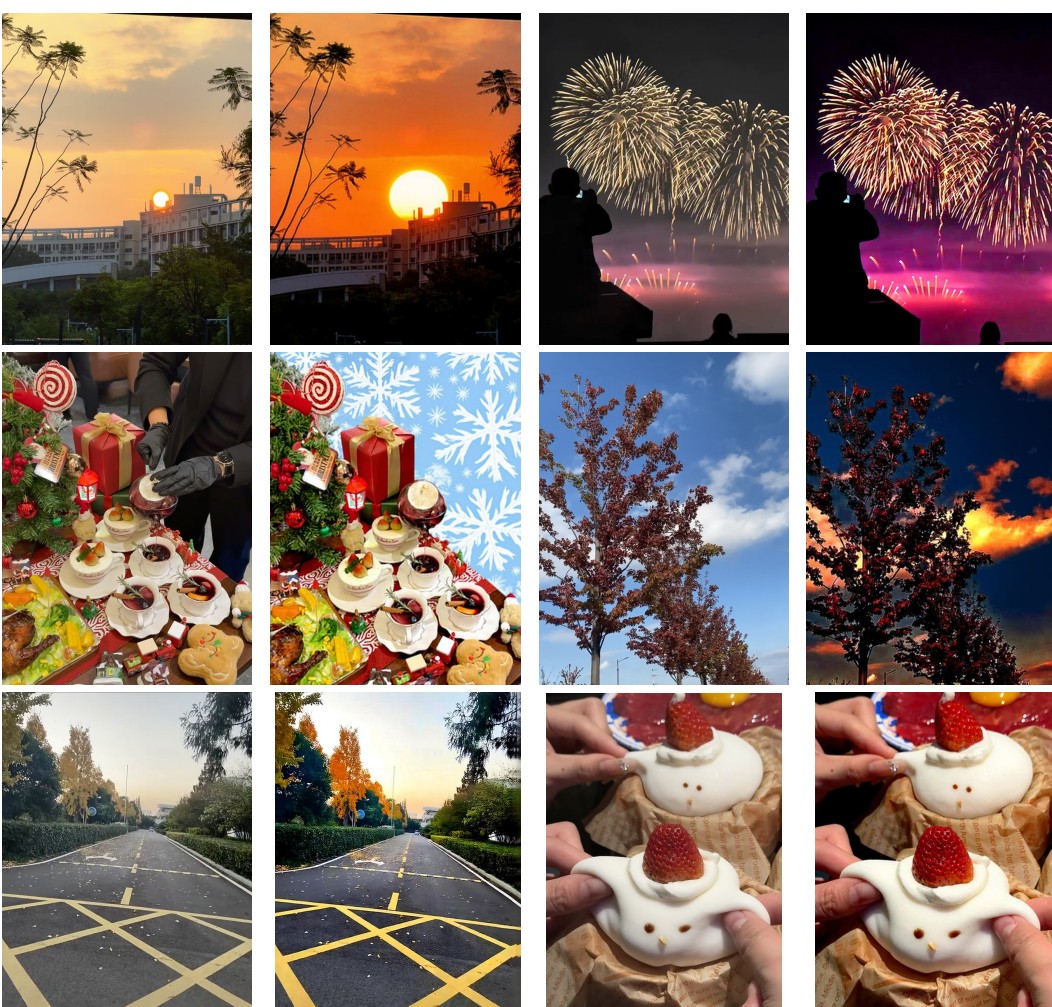

Figure 7: More visual results of PhotoAgent.

## A.4 USER STUDY ON REAL-WORLD EDITING SCENARIOS

To further validate the effectiveness and robustness of PhotoAgent, we conduct a user study involving 30 participants across 20 real-world editing scenarios, collecting a total of 600 votes. Participants are asked to select their preferred result based on both visual quality and willingness to share. As shown in Table 7, PhotoAgent is consistently favored over several baseline methods, demonstrating its effectiveness in real-world settings.

## A.5 EXPERIMENTAL DETAILS

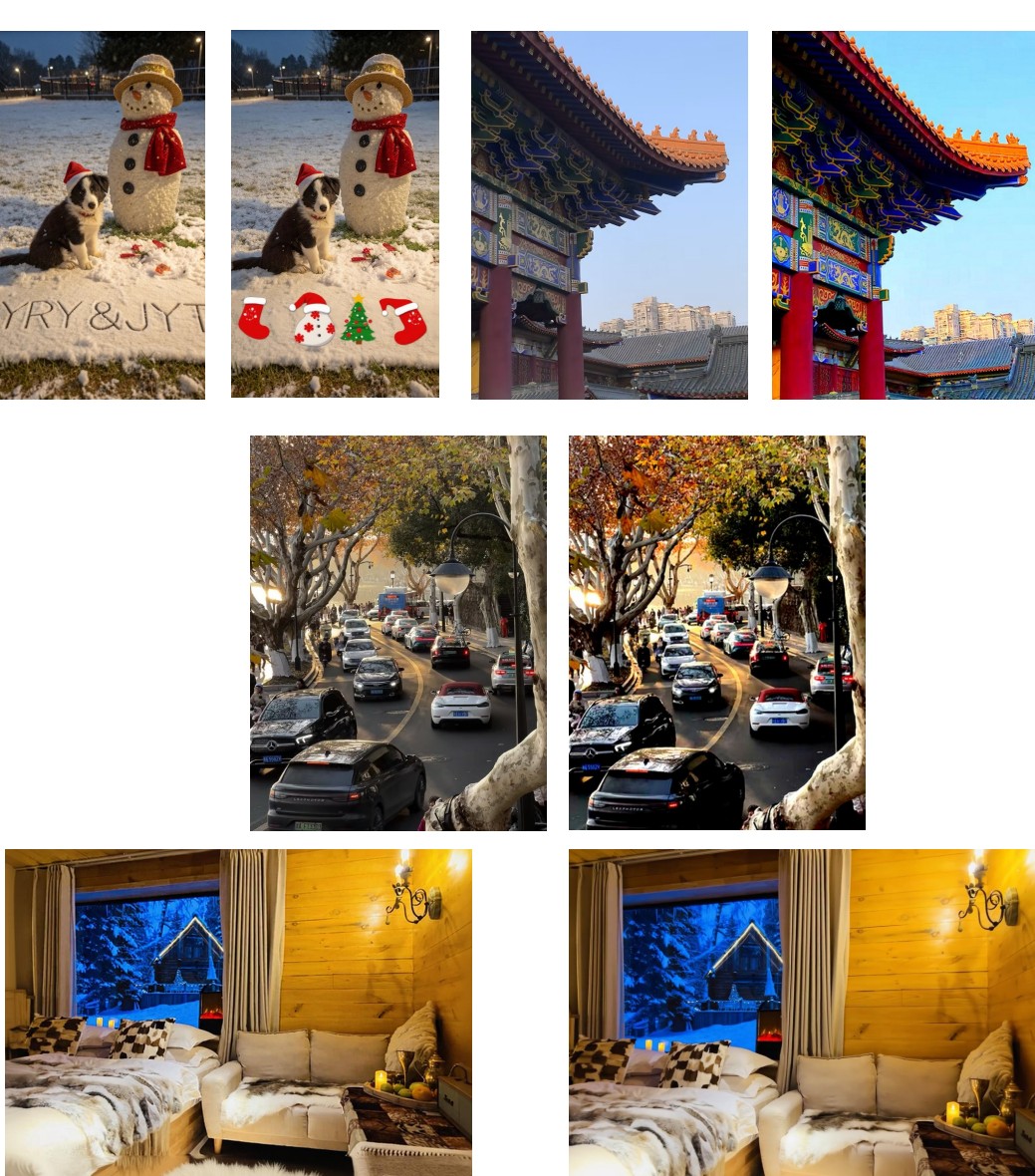

Figure 8: More visual results of PhotoAgent.

Table 7: User study results: percentage of votes selecting each method as the best.

| HuggingGPT | ReAct (cls.) | GPT-4o | PhotoAgent (Ours) |
|------------|--------------|--------|-------------------|
| 8.1% | 12.0% | 20.7% | 59.2% |

**Early Stopping.** PhotoAgent employs two complementary early stopping strategies to prevent unnecessary edits on high-quality images: (1) a maximum iteration limit, which forces termination after a predefined number of steps, and (2) a no-improvement criterion, which stops editing if the evaluator detects no significant score improvement over N consecutive iterations. These mechanisms ensure that high-quality images are not over-edited.

**Details of MCTS** Algorithm 1 shows the pseudo-code for the Monte Carlo Tree Search (MCTS) planner at the core of PhotoAgent. The search starts from the current image state $s_t$ and runs for

a set number of simulations. Each simulation follows four main phases: Selection, Expansion, Simulation (including Evaluation), and Backpropagation. Selection: The algorithm moves from the root node through the tree, choosing actions that balance exploration and exploitation using the UCT policy. This process continues until it reaches a leaf node that has not been fully expanded. Expansion: At a non-terminal leaf node $s_L$, the perceiver generates candidate editing actions. Each action corresponds to a new child node, which is added to the search tree to represent the resulting image state. Simulation: From the expanded node, the algorithm performs a lightweight rollout up to a maximum depth $d$. During this rollout, the evaluator assesses the resulting state $s_T$ and assigns a reward $G$, which reflects the predicted aesthetic and semantic quality of the edits. Backpropagation: The reward $G$ is propagated backward along the path that was traversed. This updates the visit counts $N(s, a)$ and average rewards $Q(s, a)$ for all visited nodes, helping the selection phase make better decisions in future simulations. After completing all simulations, the action from the root node with the highest visit count is chosen for execution. This action represents the most thoroughly explored and promising option.

This MCTS process enables exploratory visual aesthetic planning. By simulating multiple possible future trajectories in a fast-approximation environment, the agent can obtain the outcomes of different editing strategies without performing costly real edits. Integrating the reward model ensures that the search favors edits aligned with human preferences. As a result, the system can find high-quality actions and handle multi-step editing.

We show more visual results in Fig. 7 and Fig.8.

## A.6 DATASET DIVERSITY AND FAIRNESS

The UGC-Edit dataset is constructed from two primary sources: LAION, which is predominantly English-dominant and collected from websites, and RealQA, a Chinese-dominant dataset collected from AutoNavi. Together, these sources provide a broad coverage of real-world scenarios, including tourist attractions, restaurants, hotels, leisure venues, and other user-active locations. This diversity enables the reward model to be trained and evaluated across a variety of cultural and content contexts.

Preliminary checks on model outputs across these diverse contexts indicate no systematic bias against non-Western or unconventional aesthetic styles. We also acknowledge that certain underrepresented groups may remain, and expanding the datasets with more geographically and culturally diverse sources represents an important future direction to further enhance generalization and fairness.

## A.7 FUTURE WORK

In this section, we focus on discussing how to extend the current system to other application domains. First, different application domains may require specialized editing tools. For example, medical or scientific images often rely on stable reconstruction models rather than generative editors, so integrating more deterministic editors, such as fidelity-oriented restoration models (Potlapalli et al., 2023; Conde et al., 2024), could improve stability. Second, practical deployment frequently involves combining heterogeneous tools, including local enhancement models or commercial APIs. Creating a unified, plugin-style interface would simplify management and reduce system overhead. Third, different domains require evaluators aligned with their specific attributes, such as diagnostic or structural metrics for scientific imagery. Domain-specific reward models can help maintain consistent performance across diverse tasks. Finally, new domains often require specialized evaluation metrics. For instance, non-photorealistic or artistic images, such as illustrations, anime, or heavily stylized renderings, may need training or fine-tuning on domain-specific data. Incorporating these components would allow PhotoAgent to guide edits more effectively and make it applicable to specialized tasks.

## A.8 THE USE OF LARGE LANGUAGE MODELS

We clarify that large language models are employed only for language refinement and do not contribute to ideation, methodological design, or other creative aspects of this work.

---

**Algorithm 1** MCTS Planning for PhotoAgent

---

**Require:** Current state $s_t$, Perceiver, Executer, Evaluator, Rollout depth $d$
**Ensure:** Best action $a_{\text{best}}$

1: **while** within computational budget **do**
2:     $s \leftarrow s_t$                     ▷ Start from root state

3:     **// 1. Selection**
4:     **while** $s$ is in the tree and not terminal **do**
5:         $a \leftarrow$ select action using UCT policy from $s$
6:         $s \leftarrow$ next state after taking action $a$
7:     **end while**
8:     $s_L \leftarrow s$                   ▷ Reached leaf node $s_L$

9:     **// 2. Expansion**
10:    **if** $s_L$ is non-terminal **then**
11:        Actions $\leftarrow$ Perceiver($s_L$)
12:        Add child nodes for each action to the tree
13:    **end if**

14:    **// 3. Simulation**
15:    $s_T \leftarrow s_L$
16:    **for** $i = 1$ to $d$ **do**
17:        $a \leftarrow$ select random action from $s_T$
18:        $s_T \leftarrow$ next state after action $a$
19:        **if** $s_T$ is terminal **then**
20:            **break**
21:        **end if**
22:    **end for**
23:    $G \leftarrow$ Evaluator($s_T$)          ▷ Assess aesthetic and semantic quality

24:    **// 4. Backpropagation**
25:    Backpropagate $G$ along the path from $s_t$ to $s_L$
26:    Update $Q(s, a)$ and $N(s, a)$ for all visited nodes
27: **end while**

28: **return** $a_{\text{best}} = \arg\max_a N(s_t, a)$       ▷ Choose the most visited action

---

