# OpenReview forum: "PhotoAgent: Exploratory Visual Aesthetic Planning with Large Vision Models"
_ICLR.cc/2026/Conference — Submitted to ICLR 2026_

### Official Review · Reviewer_SjC9 · 2025-10-27

**Soundness:** 2
**Presentation:** 4
**Contribution:** 3
**Rating:** 4
**Confidence:** 3

**Summary:**

This paper proposes PhotoAgent, an autonomous system for closed-loop, reasoning-based image editing. The method reframes image enhancement as a sequential decision-making problem, integrating four modules. The authors also introduce UGC-Edit, a user-generated-content (UGC) photo preference dataset, built from LAION and RealQA with LLM-based filtering and human verification. Experiments compare PhotoAgent against single-step and agent-based baselines across multiple metrics. PhotoAgent achieves superior quantitative and qualitative results.

**Strengths:**

- Clear motivation and framework: The paper articulates the gap between user-driven and agentic editing paradigms, emphasizing the bottleneck of user-in-the-loop instruction decomposition. It convincingly positions PhotoAgent as a next step in computational photography by automating perception, reasoning, and actuation through a closed-loop design.
- Innovative integration of MCTS planning with vision-language reasoning: The combination of LLaVA-based perception, MCTS exploration, and learned reward evaluation represents a principled synthesis of planning and generation.

**Weaknesses:**

- Dataset diversity and sensitivity underexplored: Although the UGC-Edit dataset is claimed to contain “7k real photos,” it is largely filtered from LAION and RealQA, both web-based and English-dominant. There is no quantitative breakdown of geographic, cultural, or device diversity, nor any audit of sensitive attributes (e.g., faces, regions, or socioeconomic bias). Given that “visual aesthetics” vary culturally, the current dataset risks encoding narrow stylistic norms. No sensitivity analysis examines whether the reward model penalizes unconventional or non-Western aesthetics.
- Computational cost and efficiency trade-offs: The reported runtime (~580 s per image) suggests substantial overhead. Although justified by exploration depth, there is no efficiency–quality curve, and comparisons against lighter hierarchical planners or speculative rollouts are missing. For a claimed “autonomous system,” such runtime may hinder real-world usability.
- Evaluation metrics limited to aesthetic alignment: The study emphasizes perceptual quality (CLIP, Laion-Reward, BRISQUE) but lacks diversity in task scenarios. Real editing often includes semantic modification (object addition/removal, spatial relayout). The benchmarks limit the generality of the conclusions.
- Evaluator–planner coupling and circularity concerns: The learned reward model both trains on and evaluates UGC-style content. While ablations remove the evaluator to show degradation, there is no external validation on unseen evaluators or human-in-the-loop scoring beyond the correlation table. PhotoAgent may optimize toward its own learned aesthetic metric rather than general human preferences

**Questions:**

- Is the learned evaluator robust to out-of-distribution edits or artistic styles beyond photographic realism?

---

> ### Author Response · Authors · 2025-11-26
> **Response to Reviewer SjC9 (1/2)**
>
> We sincerely thank the reviewer for the insightful comments. We revise our paper according to the reviewer's comments. We hope our response below can address the reviewer's concerns.
>
>
> > **W1**  Dataset diversity and sensitivity underexplored: Although the UGC-Edit dataset is claimed to contain “7k real photos,” it is largely filtered from LAION and RealQA, both web-based and English-dominant. There is no quantitative breakdown of geographic, cultural, or device diversity, nor any audit of sensitive attributes (e.g., faces, regions, or socioeconomic bias). Given that “visual aesthetics” vary culturally, the current dataset risks encoding narrow stylistic norms. No sensitivity analysis examines whether the reward model penalizes unconventional or non-Western aesthetics.
>
>
> Thank you for the valuable comment. We provide further details on our dataset composition. LAION is primarily sourced from **English-dominant** content, while RealQA is a **Chinese-dominant** dataset collected from AutoNavi. We note that the datasets cover a wide range of real-world scenarios, such as tourist attractions, restaurants, hotels, leisure venues, and other user-active locations, allowing our reward model to be trained and evaluated across diverse cultural and content contexts. Preliminary checks on model outputs across these diverse content contexts show no systematic bias against non-Western or unconventional aesthetics. We also acknowledge that some underrepresented groups may remain. Expanding the datasets with more diverse sources and cultural contexts represents an important direction for future work to further improve generalization and fairness. We revise the paper and put these discussions in Appendix A.6.
>
>
> > **W2**  Computational cost and efficiency trade-offs: The reported runtime (~580 s per image) suggests substantial overhead. Although justified by exploration depth, there is no efficiency–quality curve, and comparisons against lighter hierarchical planners or speculative rollouts are missing. For a claimed “autonomous system,” such runtime may hinder real-world usability.
>
> Thank you for the valuable comment. To better understand the computational cost and scalability of PhotoAgent, we conducted a detailed profiling at \~1080p (1440*1080) resolution to identify the main runtime bottlenecks. As shown in the following table, MCTS simulations dominate the runtime (\~53%), Executor accounts for \~38%, and other components such as the Perceiver and Evaluator contribute smaller portions. Note that the multiple (20 times) MCTS simulations make its measured execution time longer than in actual deployment. This profiling provides clear guidance for targeted optimization.
>
>
> |Component|Time(s)|Percentage(total/parent)|
> |-|-:|-|
> |Perceiver|~10|2.1%|
> |Planner(MCTS)|~250|53.2%/100%|
> |├─*Executor(MCTS)*|~170|36.2%/68.0%|
> |└─*Evaluator(MCTS)*|~80|17.0%/32.0%|
> |Executor|~180|38.3%|
> |Evaluator|~30|6.4%|
> |Total*|~470|100%|
>
>
> Based on the above analysis, we outline three concrete acceleration directions.
> (1) As shown in the following table, reducing the number of MCTS simulations can substantially speed up the system, with only a marginal performance drop.
>
>
> |Simulations|Time(s)|BRISQUE(↓)|Laion Reward(↑)|UGC score(↑)|
> |-:|-:|-:|-:|-:|
> |5|~60|0.6723| 0.4816| 3.217|
> |10|~120|0.6522| 0.4865| 3.240|
> |15|~185|0.6429| 0.4830| 3.368|
> |20|~250|0.6103| 0.4934| 3.465|
>
> (2) PhotoAgent can readily benefit from faster editing models, where different tools may call different times [1]. For example, at the same resolution (1440p), Step1x-Edit requires only about half the runtime of Flux.1 Kontext-Dev (reducing the time from  ~20s to ~10s) on two Nvidia A800 GPUs. Only minimal changes to the API call and model selection are needed, and the rapid development of editing models offers clear potential for further improvement.
>
> (3) In production-level deployments, using more efficient generative and evaluation models, such as FP8/FP4 quantization or TensorRT optimization, can substantially reduces latency. For example, quantizing FLUX.1 Kontext’s transformer, which contributes roughly 96% of its computation, to FP8 or FP4 yields over 2x memory savings and noticeably faster inference on NVIDIA Blackwell GPUs [2]. We update the paper and include these analyses in Appendix A.1.

---

> ### Author Response · Authors · 2025-11-26
> **Response to Reviewer SjC9 (2/2)**
>
> > **W3**  Evaluation metrics limited to aesthetic alignment: The study emphasizes perceptual quality (CLIP, Laion-Reward, BRISQUE) but lacks diversity in task scenarios. Real editing often includes semantic modification (object addition/removal, spatial relayout). The benchmarks limit the generality of the conclusions.
>
>
> Thank you for the valuable comment. We note that conducting a direct semantic-level alignment evaluation is inherently challenging, as PhotoAgent operates as a fully autonomous agent given only a high-level instruction to enhance image aesthetics, without detailed user guidance. To address this, in our user study, participants were explicitly asked to consider both **willingness to share** and whether the edited image deviates excessively from the original when selecting preferred results. This assesses the semantic reliability of edits, ensuring modifications are coherent and meaningful. The results show that the majority of participants considered PhotoAgent’s edits appropriate, further supporting its semantic reliability (**Appendix A.4**). We agree that more targeted semantic evaluation would strengthen generalization assessment and consider this an important direction for future work.
>
> > **W4**  Evaluator–planner coupling and circularity concerns: The learned reward model both trains on and evaluates UGC-style content. While ablations remove the evaluator to show degradation, there is no external validation on unseen evaluators or human-in-the-loop scoring beyond the correlation table. PhotoAgent may optimize toward its own learned aesthetic metric rather than general human preferences.
>
> Thanks for raising this important point. To address this, we conducted an additional human preference study comparing PhotoAgent with single-step editors and agent-based planners. In total, 600 votes were collected in a user study on real-world editing, involving 30 participants across 20 scenarios. Participants selected their preferred results based on both visual quality and willingness to share. As shown in the following table, the system aligns with general human perception, not just the learned reward. We revise the paper and put these details in Appendix A.4.
>
> |HuggingGPT|ReAct(cls.)|GPT-4o|Ours|
> |-|-|-|-|
> |8.1%|12.0%|20.7%|59.2%|
>
>
> > **Q1**  Is the learned evaluator robust to out-of-distribution edits or artistic styles beyond photographic realism?
>
> Thank you for pointing out this valuable research direction. To test generalization, we evaluate our reward model on an external UGC dataset PARA [1] with diverse content and lighting conditions. As shown in the following table (**Appendix A.3**), our model shows high correlation with aesthetic judgments, demonstrating its generalization ability (SRCC~0.75, surpassing previous state-of-the-art PIAA model [2] of ~0.70–0.72). Moreover, in our implementation, we use multiple evaluators for assessment, which substantially enhances the robustness of aesthetic judgments. Regarding artistic testing, since the evaluator is trained primarily on user-generated photographs, a comprehensive evaluation on non-photorealistic or artistic styles is beyond the current scope, which would be a potential direction to explore. We have revised the paper and included these discussions in Appendix A.3.
>
> |Metric|Aesthetic|Content|
> |-|-|-|
> |PLCC|0.7390|0.7577|
> |SRCC|0.7560|0.7702|
>
>
> > Reference
>
> [1] https://developer.nvidia.com/blog
>
> [2] Personalized Image Aesthetics Assessment with Rich Attributes, CVPR, 2022.
>
> [3] Personalized Image Aesthetics Assessment with Attribute-guided Fine-grained Feature Representation, ACM MultiMedia, 2022.

---

### Official Review · Reviewer_Vyiu · 2025-10-28

**Soundness:** 3
**Presentation:** 3
**Contribution:** 3
**Rating:** 6
**Confidence:** 3

**Summary:**

1. PhotoAgent is introduced as an autonomous image editing system that reframes image enhancement as a sequential decision-making process, reducing reliance on detailed step-by-step prompt engineering from users.

2. The framework integrates large language models for intentional reasoning and dynamic planning with vision-language models for precise localized editing, forming a closed-loop process that iteratively refines outputs based on visual feedback.

3. Its workflow involves a Perception & Planner to propose candidate actions, MCTS search to explore and prune editing sequences, an Executor to apply the selected actions, and an Evaluator to score intermediate results and trigger re-planning when necessary, enabling robust multi-step image editing.

4. The authors further contribute the UGC-Edit preference dataset, a large-scale collection of multi-step editing tasks based on real user photos, along with a reward model trained on this dataset for evaluating visual quality.

5. Extensive experiments show that PhotoAgent outperforms existing methods in semantic faithfulness and visual quality across diverse editing scenarios.

**Strengths:**

1. The paper introduces a clear reframing of image editing as a sequential decision-making problem, reducing reliance on detailed step-by-step prompt engineering and enabling autonomous exploratory visual aesthetic planning.

2.The closed-loop integration of large language models for intentional reasoning and dynamic planning with vision-language models for precise localized editing is well-motivated and convincingly presented as a holistic pipeline.

3. The system demonstrates strong empirical performance: PhotoAgent consistently outperforms baselines across semantic alignment and non-reference aesthetic quality metrics such as CLIP Similarity, LPIPS, BRISQUE, Laion-Reward, and UGC Score.

4. The introduction of the UGC-Edit preference dataset and reward model is a valuable contribution, providing a benchmark and evaluation mechanism that better captures user-generated content preferences.

**Weaknesses:**

1. While the system design is interesting, the paper leans more towards an engineering integration of existing models rather than introducing fundamentally novel algorithmic insights. This may limit its perceived contribution.

2. The evaluation, although comprehensive in metrics, could be strengthened with more diverse baselines or user studies in real-world editing scenarios to better validate practical utility and robustness.

3. Some technical descriptions, such as the role of the “controller” (in the caption of Fig. 1) remain under-specified. Clarifying these components would improve the reproducibility and clarity of the work.

**Questions:**

1. Could the authors clarify how their approach differs from prior work that also leverages large vision models for aesthetic evaluation and editing? At present, the novelty relative to earlier systems is not fully clear.

2. The paper briefly mentions a “controller” (in the caption of Fig. 1) that uses MCTS, but this concept is only introduced once and not elaborated further. Could the authors provide more details on the controller’s role and implementation?

3. How well does the proposed framework generalize to broader editing tasks or domains beyond the photo scenarios tested? It would be helpful to understand potential limitations in applicability.

---

> ### Author Response · Authors · 2025-11-26
> **Response to Reviewer Vyiu (1/2)**
>
> We sincerely thank the reviewer for the positive assessment of our work and insightful comments. We revise our paper according to the reviewer's comments. We hope our response below can address the reviewer's concerns.
>
> > **W1**  While the system design is interesting, the paper leans more towards an engineering integration of existing models rather than introducing fundamentally novel algorithmic insights. This may limit its perceived contribution.
>
> We appreciate the reviewer's ackownledgement on our system design. We would like to clarify that PhotoAgent represents a **system-level** innovation, not merely an integration of existing models. The key novelty and value arise from the **coordinated interaction of multiple modules**, which enables emergent agent-level decision-making that cannot be achieved by any component in isolation.
>
> PhotoAgent’s key technical contributions at the system level include: (1) it integrates exploratory multi-step planning via MCTS to anticipate long-term effects of edits, (2) it employs a task-specific reward model trained on real user-generated content to provide human-aligned guidance, (3) it coordinates multiple editing tools in a closed-loop framework to prevent error accumulation, and (4) it enables emergent agent-level decision-making that cannot be achieved by any single component alone.
>
> We will also **release the code and dataset** to the community to facilitate further research and enable broader adoption. We revise the Introduction and Conclusion to further emphasized our novelty.
>
>
>
> > **W2**  The evaluation, although comprehensive in metrics, could be strengthened with more diverse baselines or user studies in real-world editing scenarios to better validate practical utility and robustness.
>
> Thank you for the valuable comment. We provide experiments with user studies in real-world editing. In total, 600 votes were collected in a user study on real-world editing, involving 30 participants across 20 scenarios. Participants selected their preferred results based on both visual quality and willingness to share. As shown in the following table, **PhotoAgent is consistently favored**, demonstrating strong practical utility and robustness. We revise the paper and put these details in Appendix A.4.
>
> |HuggingGPT|ReAct(cls.)|GPT-4o|Ours|
> |-|-|-|-|
> |8.1%|12.0%|20.7%|59.2%|
>
>
>
> > **W3**  Some technical descriptions, such as the role of the “controller” (in the caption of Fig. 1) remain under-specified. Clarifying these components would improve the reproducibility and clarity of the work.
> > **Q2**  The paper briefly mentions a “controller” (in the caption of Fig. 1) that uses MCTS, but this concept is only introduced once and not elaborated further. Could the authors provide more details on the controller’s role and implementation?
>
> In our system, the controller is an abstract, rule-based module that manages the entire editing loop. Specifically, the controller's primary responsibilities are to:
>
> 1. Oversee the iterative planning, execution, and evaluation workflow.
> 2. Manage the overall editing state  (e.g., current image, action history, search statistics).
> 3. Determine termination based on convergence or step limits.
> 4. Forward information to MCTS and trigger operations when conditions are met (e.g., switching tools or adjusting planning).
>
> This design **keeps the system modular and extensible** without altering core logic. We have updated the figure caption and Section 3 to clearly clarify this design.

---

> ### Author Response · Authors · 2025-11-26
> **Response to Reviewer Vyiu (2/2)**
>
> > **Q1**  Could the authors clarify how their approach differs from prior work that also leverages large vision models for aesthetic evaluation and editing? At present, the novelty relative to earlier systems is not fully clear.
>
>
> Thank you for the question. Our contributions differ from both prior instruction-based editors and existing agent-based editing systems in the following ways:
>
> **(1) Compared with prior image editing models using large vision models (e.g., InstructPix2Pix, SDXL, GPT-4o, Flux, Bagel):**
>
> These models function as *single-step, user-in-the-loop* editors and depend on users to design precise sequential prompts and iterate manually. PhotoAgent instead **automates the entire decision** process by perceiving the image, planning multi-step edits, selecting tools, and evaluating intermediate results. This shifts the workflow from prompt engineering toward autonomous long-horizon editing.
>
>
> **(2) Compared with existing agent-based systems (e.g., JarvisArt, 4KAgent):**
> Prior agents mainly focus on *low-level adjustments* (color, contrast, illumination) and follow an *short-term*, fixed action sequence. PhotoAgent instead performs **long-term closed-loop semantic planning** with an aesthetic evaluator and MCTS, enabling coherent, high-level enhancements and avoiding irreversible errors.
>
> We revise the Introduction and Related Work to further emphasized these contributions.
>
>
> > **Q3**  How well does the proposed framework generalize to broader editing tasks or domains beyond the photo scenarios tested? It would be helpful to understand potential limitations in applicability.
>
>
>
> Thank you for pointing out this direction worthy of further analysis. While our experiments focus on photo editing, the **PhotoAgent framework is designed to be modular and general**, allowing it to be adapted to broader tasks and domains. Potential applications include medical imaging enhancement, satellite and remote sensing image processing, and scientific visualization. In each case, task-specific editors can replace the photo editor, and appropriate reward models can guide high-quality edits.
>
>
> At the same time, we provide potential directions that could further enhance the generalization. **First**, new domains require compatible editing tools. For example, medical or scientific imaging often depends on stable reconstruction models rather than generative editors. Using more deterministic editors such as fidelity-oriented restoration models [1,2] can improve stability. **Second**, practical deployment requires integrating heterogeneous tools, including local enhancement models or commercial APIs. A unified plugin-style interface can streamline management and reduce overhead. **Third**, different domains require evaluators that match their specific semantics, such as diagnostic or structural metrics for scientific imagery. Adopting domain-specific reward models can help sustain performance across domains. We revise the paper and add these discussions in Appendix A.7.
>
>
> > Reference
>
> [1] PromptIR: Prompting for All-in-One Image Restoration, NeurIPS, 2023.
> [2] InstructIR: High-Quality Image Restoration Following Human Instructions, ECCV, 2024

---

> > ### Comment · Reviewer_Vyiu · 2025-11-27
> > **Response to author**
> >
> > Thank you for your further clarification. I think almost all of my concerns have been addressed. At this stage, I am inclined to keep the current positive rating

---

### Official Review · Reviewer_CZkY · 2025-10-29

**Soundness:** 3
**Presentation:** 3
**Contribution:** 3
**Rating:** 6
**Confidence:** 5

**Summary:**

This paper presents PhotoAgent, an autonomous image editing system that addresses the limitation of current instruction-based editing models requiring detailed, expert-level prompts. The system employs a closed-loop architecture consisting of four components: (1) a VLM-based perceiver that generates candidate editing actions, (2) an MCTS-based planner that explores multi-step editing trajectories, (3) an executor that applies edits using both classical and generative tools, and (4) an evaluator powered by a custom reward model. To support this work, the authors construct the UGC-Edit dataset containing 7K+ user-generated content images with editing intents, and train a Qwen2.5-VL-based reward model via Group Relative Policy Optimization (GRPO) to align with human aesthetic preferences. Experiments demonstrate that PhotoAgent outperforms single-step baselines and other agent-based approaches on semantic alignment and aesthetic quality metrics when given ambiguous editing instructions like "make this image better."

**Strengths:**

1、The UGC-Edit dataset addresses a gap in existing benchmarks by focusing on realistic user photo editing scenarios. The construction process with LLM-based filtering and human verification appears rigorous.

2、Training a specialized reward model via GRPO that achieves strong correlation with human judgments is a good contribution. The validation against baseline metrics demonstrates its superiority in capturing aesthetic preferences.

3、 The paper includes both quantitative metrics (CLIP, BRISQUE, LPIPS, Laion-Reward, UGC Score) and qualitative comparisons across diverse baselines, with thorough ablation studies validating key design choices.

**Weaknesses:**

1、 While the system integration is effective, the individual components (VLM perception, MCTS planning, reward model training via GRPO) are relatively standard techniques.

2、 The paper mentions computational expense as a limitation but provides no concrete analysis of runtime, cost per image, or number of MCTS simulations needed. Given the complexity of the approach (MCTS with generative model rollouts), scalability and practical deployment concerns are significant but unexplored.

3、The lack of quantitative comparisons with relevant work is notable. While the authors mention recent studies using agentic models for image editing, they appear to have omitted quantitative metric comparisons. Could the authors briefly compare their approach with existing works such as JarvisArt[1] and MonetGPT[2]?

[1] [NeurIPS' 2025] JarvisArt: Liberating Human Artistic Creativity via an Intelligent Photo Retouching Agent
[2] SIGGRAPH 2025 MonetGPT: Solving Puzzles Enhances MLLMs' Image Retouching Skills

**Questions:**

see Weaknesses

---

> ### Author Response · Authors · 2025-11-26
> **Response to Reviewer CZkY (1/2)**
>
> We sincerely thank the reviewer for the positive assessment of our work and insightful comments. We revise our paper according to the reviewer's comments. We hope our response below can address the reviewer's concerns.
>
> > **W1** While the system integration is effective, the individual components (VLM perception, MCTS planning, reward model training via GRPO) are relatively standard techniques.
>
> Thank you for acknowledging our system's effectiveness. We would like to further clarify that PhotoAgent is a **system-level** innovation, not merely an integration of existing modules. The novelty arises from the **coordinated interaction of multiple modules**, which enables emergent agent-level decision-making that cannot be achieved by any component in isolation, effectively mimicking human expert reasoning.
>
> PhotoAgent’s key technical contributions at the system level include: (1) it integrates exploratory multi-step planning via MCTS to anticipate long-term effects of edits, (2) it employs a task-specific reward model trained on real user-generated content to provide human-aligned guidance, (3) it coordinates multiple editing tools in a closed-loop framework to prevent error accumulation, and (4) it enables emergent agent-level decision-making that cannot be achieved by any single component alone.
>
> We will also **release the code and dataset** to the community to facilitate further research and enable broader adoption. We revise the Introduction and Conclusion to provide a clear novelty claim.
>
> > **W2**  The paper mentions computational expense as a limitation but provides no concrete analysis of runtime, cost per image, or number of MCTS simulations needed. Given the complexity of the approach (MCTS with generadtive model rollouts), scalability and practical deployment concerns are significant but unexplored.
>
> Thank you for the valuable comment. To better understand the computational cost and scalability of PhotoAgent, we conducted a detailed profiling at \~1080p (1440*1080) resolution to identify the main runtime bottlenecks. As shown in the following table, MCTS simulations dominate the runtime (\~53%), Executor accounts for \~38%, and other components such as the Perceiver and Evaluator contribute smaller portions. Note that the multiple (20 times) MCTS simulations make its measured execution time longer than in real environment. This profiling provides clear guidance for targeted optimization.
>
>
> |Component|Time(s)|Percentage(total/parent)|
> |-|-:|-|
> |Perceiver|~10|2.1%|
> |Planner(MCTS)|~250|53.2%/100%|
> |├─*Executor(MCTS)*|~170|36.2%/68.0%|
> |└─*Evaluator(MCTS)*|~80|17.0%/32.0%|
> |Executor|~180|38.3%|
> |Evaluator|~30|6.4%|
> |Total*|~470|100%|
>
> *Total\* excludes the 30s model-initialization overhead and the duplicated 80s evaluator time that was previously counted within the planner, yielding the corrected runtime.*
>
>
> Based on these insights, we explore three strategies to improve efficiency:
>
> (1) Reducing the number of MCTS simulations. Table 2 shows that fewer simulations substantially decreases runtime while only slightly affecting performance. This indicates that runtime can be tuned to meet practical requirements without major quality loss.
>
>
> |Simulations|Time(s)|BRISQUE(↓)|Laion Reward(↑)|UGC score(↑)|
> |-:|-:|-:|-:|-:|
> |5|~60|0.6723| 0.4816| 3.217|
> |10|~120|0.6522| 0.4865| 3.240|
> |15|~185|0.6429| 0.4830| 3.368|
> |20|~250|0.6103| 0.4934| 3.465|
>
>
> (2) Faster models can be easily swapped with minimal API changes, further reducing processing time without compromising quality. For example, at the same resolution (1080p), Step1x-Edit requires only about half the runtime of Flux.1 Kontext-Dev (reducing the time from ~20s to ~10s).
>
>
> (3) In production-level deployments, memory and inference time can be further optimized using techniques such as FP8/FP4 quantization and GPU-specific optimizations (e.g., TensorRT). For example, quantizing the transformer backbone in Flux.1 Kontext could achieve over 2× memory savings and noticeably faster runtime on NVIDIA Blackwell GPUs [1], enabling more practical large-scale editing without compromising performance.
>
> We add these discussion in Appendix A.1 of the revised paper.

---

> ### Author Response · Authors · 2025-11-26
> **Response to Reviewer CZkY (2/2)**
>
> > **W3** The lack of quantitative comparisons with relevant work is notable. While the authors mention recent studies using agentic models for image editing, they appear to have omitted quantitative metric comparisons. Could the authors briefly compare their approach with existing works such as JarvisArt[1] and MonetGPT[2]?[1] [NeurIPS' 2025] JarvisArt: Liberating Human Artistic Creativity via an Intelligent Photo Retouching Agent [2] SIGGRAPH 2025 MonetGPT: Solving Puzzles Enhances MLLMs' Image Retouching Skills
>
>
> Thank you for the valuable comment. First, we would like to clarify that unlike JarvisArt and MonetGPT, which are limited to **pure retouching** editing (e.g., adjusting color, tone, or exposure via procedural software tools, such as Lightroom or GIMP), PhotoAgent targets a **more general and semantically meaningful level** of editing, allowing programmatic control through flexible APIs and open-source editing platforms (e.g., adding a sun to a dim sky or modifying objects within the scene, **Fig. 7 in Appendix**). Moreover, we provide quantitative comparisons with several agent-based methods (e.g., HuggingGPT and ReAct) in Table 1, demonstrating that PhotoAgent performs competitively across multiple metrics. We include these discussions in Introduction and Related Work and will further investigate applying our framework to image retouching in future work.
>
>
> > Reference
>
> [1] https://developer.nvidia.com/blog

---

### Official Review · Reviewer_MNoE · 2025-10-30

**Soundness:** 3
**Presentation:** 4
**Contribution:** 4
**Rating:** 6
**Confidence:** 3

**Summary:**

This paper addresses the pain point that existing instruction-based image editing models heavily rely on users to design precise, multi-step instructions. It proposes an autonomous image editing agent framework called PhotoAgent. The system aims to simulate the decision-making process of a human expert, automatically performing complex image aesthetic enhancement tasks without requiring detailed step-by-step prompts from the user. At its core is a closed-loop "perceive-plan-execute-evaluate" process. The paper also contributes a new dataset called the UGC-Edit Preference Dataset and trains a new reward model (UGC Evaluator) based on it. Experiments show this model has a very high correlation with human preferences.

**Strengths:**

This paper redefines image editing as an "exploratory visual aesthetic planning" problem. By introducing forward-looking planning through the MCTS algorithm , the agent can evaluate the long-term impact of multi-step editing sequences. This avoids the local optima or "short-sighted" decisions that greedy methods (like ReAct (Closed-loop)) are prone to.

The paper emphasizes the central role of the evaluator in a closed-loop system. Instead of relying on existing general-purpose aesthetic models that perform poorly on UGC data , the authors built the UGC-Edit dataset and trained a specialized reward model. Table 2 demonstrates that this evaluator (PLCC 0.8300) aligns far better with human preferences than existing models (e.g., Laion-aesthetic, PLCC 0.5567).


The experimental design compares against comprehensive baselines, including non-agent SOTA models (GPT-4o, Flux.1) and agent-based models (HuggingGPT, ReAct) . Given a vague instruction, PhotoAgent achieves the best results across all metrics (Table 1), robustly proving the superiority of the proposed framework. Ablation studies also clearly validate the necessity of both MCTS and the new evaluator .

**Weaknesses:**

- The most significant weakness of this paper is its extremely high computational latency. Section 5.5 mentions that under the standard configuration, the average processing time per image is approximately 580 seconds. This high time cost makes the system currently unsuitable for any practical or interactive applications, rendering it more of a "proof-of-concept."

- To make MCTS feasible, the planner relies on "reduced-resolution processing" in a "fast-approximation environment". A potential issue is that an edit that looks "good" at low resolution (i.e., high simulated reward) may not be so when executed at full resolution (e.g., it might introduce artifacts not visible at low resolution). This "sim-to-real" gap could mislead the MCTS planner.

- The UGC-Edit dataset (7k images)  is a good contribution, but its scale is relatively limited. The evaluation test set is also small (89 images). It remains to be seen whether this reward model can generalize well to broader and more diverse UGC images (e.g., different cultures, lighting conditions, or photographic styles).

**Questions:**

1. **Regarding computational cost:** In the MCTS planning phase, is the main computational bottleneck the "simulation" stage in the low-resolution approximate environment, or the "evaluation" stage when calling the reward model? Are there any timing statistics for the different stages to identify the performance bottleneck?

2. **Sim-to-Real Gap:** MCTS simulates at low resolution , while the executor operates at full resolution . Is there a significant reward difference between these two? Does the system frequently encounter situations where MCTS simulation predicts a high reward for an action sequence, but after full-resolution execution, the Evaluator gives a low score, causing the action to be rolled back?

3. **Regarding the stopping criterion:** The system may make unnecessary modifications to already high-quality images. Does this imply that the evaluator struggles to assign a saturated score for "perfect" images, causing the agent to always want to "do something more"? Is there an explicit early termination mechanism in the system? For example, does the system automatically stop when the evaluator's score reaches a certain threshold or when it fails to improve the score for several consecutive iterations?

---

> ### Author Response · Authors · 2025-11-26
> **Response to Reviewer MNoE (1/2)**
>
> We sincerely thank the reviewer for the positive assessment of our work and insightful comments. We revise our paper according to the reviewer's comments. We hope our response below can address the reviewer's concerns.
>
> > **W1** The most significant weakness of this paper is its extremely high computational latency. Section 5.5 mentions that under the standard configuration, the average processing time per image is approximately 580 seconds. This high time cost makes the system currently unsuitable for any practical or interactive applications, rendering it more of a "proof-of-concept."
>
> > **Q1** Regarding computational cost: In the MCTS planning phase, is the main computational bottleneck the "simulation" stage in the low-resolution approximate environment, or the "evaluation" stage when calling the reward model? Are there any timing statistics for the different stages to identify the performance bottleneck?
>
> Although the current runtime is relatively high, this does not limit PhotoAgent to a conceptual demonstration. The system is a fully functional and extensible framework whose effectiveness is determined by autonomous editing rather than interactive speed. Autonomous photo enhancement does not require real-time operation, and in many professional workflows including those involving human experts, high-quality editing naturally involves substantial computation.
>
> To better understand the computational bottleneck, we profiled PhotoAgent. As shown in the following table, most of the cost arises from MCTS planning, where simulation and in-loop execution dominate the runtime (\~53%), followed by executor calls(\~38%). Note that the multiple (20 times) MCTS simulations make its measured execution time longer than in real environment. This profiling provides clear guidance for targeted optimization.
>
> |Component|Time(s)|Percentage(total/parent)|
> |-|-:|-|
> |Perceiver|~10|2.1%|
> |Planner(MCTS)|~250|53.2%/100%|
> |├─*Executor(MCTS)*|~170|36.2%/68.0%|
> |└─*Evaluator(MCTS)*|~80|17.0%/32.0%|
> |Executor|~180|38.3%|
> |Evaluator|~30|6.4%|
> |Total*|~470|100%|
>
> *Total\* excludes the 30s model-initialization overhead and the duplicated 80s evaluator time that was previously counted within the planner, yielding the corrected runtime.*
>
>
>
> Based on the above analysis, we outline three concrete acceleration directions.
> (1) As shown in the following table, reducing the number of MCTS simulations can substantially speed up the system, with only a marginal performance drop.
>
>
> |Simulations|Time(s)|BRISQUE(↓)|Laion Reward(↑)|UGC score(↑)|
> |-:|-:|-:|-:|-:|
> |5|~60|0.6723| 0.4816| 3.217|
> |10|~120|0.6522| 0.4865| 3.240|
> |15|~185|0.6429| 0.4830| 3.368|
> |20|~250|0.6103| 0.4934| 3.465|
>
> (2) PhotoAgent can readily benefit from faster editing models, where different tools may call different times [1]. For example, at the same resolution (1440p), Step1x-Edit requires only about half the runtime of Flux.1 Kontext-Dev (reducing the time from  ~20s to ~10s) on two Nvidia A800 GPUs. Only minimal changes to the API call and model selection are needed, and the rapid development of editing models offers clear potential for further improvement.
>
> (3) In production-level deployments, using more efficient generative and evaluation models, such as FP8/FP4 quantization or TensorRT optimization, can substantially reduces latency. For example, quantizing FLUX.1 Kontext’s transformer, which contributes roughly 96% of its computation, to FP8 or FP4 yields over 2x memory savings and noticeably faster inference on NVIDIA Blackwell GPUs [2].
>
> We have put these dicussions in Appendix A.1 of the revised paper.

---

> ### Author Response · Authors · 2025-11-26
> **Response to Reviewer MNoE (2/2)**
>
> > **W2** To make MCTS feasible, the planner relies on "reduced-resolution processing" in a "fast-approximation environment". A potential issue is that an edit that looks "good" at low resolution (i.e., high simulated reward) may not be so when executed at full resolution (e.g., it might introduce artifacts not visible at low resolution). This "sim-to-real" gap could mislead the MCTS planner.
>
> > **Q2** Sim-to-Real Gap: MCTS simulates at low resolution , while the executor operates at full resolution. Is there a significant reward difference between these two? Does the system frequently encounter situations where MCTS simulation predicts a high reward for an action sequence, but after full-resolution execution, the Evaluator gives a low score, causing the action to be rolled back?
>
> In the implementation, we have already taken this into consideration and the gap is minor.
>
> (1) The evaluator itself exhibits robust and consistent behavior across both the simulated and full-resolution environments. As shown in the following table, the ranking of candidate edits obtained in low-resolution simulation strongly aligns with the ranking computed in the full-resolution execution.
>
> |Metric|1/2 resolution| 1/4 resolution|
> |-|-:|-:|
> |Top-1 retention (same best)|85%|75%|
> |Top-3 retention|100%|90%|
> |Spearman rank correlation|0.94|0.79|
> |Kendall τ|0.90|0.73|
>
>
> (2) We use the **top-K candidates and re-evaluate** them at full resolution. This ensures that occasional errors in low-resolution scoring do not mislead the planner, and that the action ultimately executed is the best in the real environment.
>
>
> (3) The system executes only one MCTS-selected edit at a time. After performing this edit at full resolution, MCTS is **restarted from the updated image**. This step-by-step closed-loop process ensures that any potential discrepancy in simulation does not accumulate, keeping each decision aligned with the real environment.
>
> We add these dicussions in Appendix A.2 of the revised paper.
>
>
> > **W3** The UGC-Edit dataset (7k images) is a good contribution, but its scale is relatively limited. The evaluation test set is also small (89 images). It remains to be seen whether this reward model can generalize well to broader and more diverse UGC images (e.g., different cultures, lighting conditions, or photographic styles).
>
>
> Thank you for pointing out this valuable research direction. To test generalization, we evaluate our reward model on an external UGC dataset PARA [3] with diverse content and lighting conditions. As shown in the following table, our model shows high correlation with aesthetic judgments, demonstrating its generalization ability (SRCC~0.75, surpassing previous state-of-the-art PIAA model [4] of ~0.70–0.72). Moreover, in our implementation, we use multiple evaluators for assessment, which substantially enhances the robustness of aesthetic judgments. We add these discussions in Appendix A.3 of the revised paper.
>
> |Metric|Aesthetic|Content|
> |-|-|-|
> |PLCC|0.7390|0.7577|
> |SRCC|0.7560|0.7702|
>
> > **Q3**  Regarding the stopping criterion: The system may make unnecessary modifications to already high-quality images. Does this imply that the evaluator struggles to assign a saturated score for "perfect" images, causing the agent to always want to "do something more"? Is there an explicit early termination mechanism in the system? For example, does the system automatically stop when the evaluator's score reaches a certain threshold or when it fails to improve the score for several consecutive iterations?
>
>
> We appreciate the reviewer’s insightful observation, which is a very valuable point.    PhotoAgent employs two complementary early stopping strategies to prevent unnecessary edits on high-quality images: (1) a maximum iteration limit, which forces termination after a predefined number of steps, and (2) a no-improvement criterion, which stops editing if the evaluator detects no significant score improvement over N consecutive iterations. These mechanisms ensure that high-quality images are not over-modified. We add these details in Appendix A.5 of the revised paper.
>
>
>
> > Reference
>
> [1] https://replicate.com/pricing
>
> [2] https://developer.nvidia.com/blogoptimizing-flux-1-kontext-for-image-editing-with-low-precision-quantization
>
> [3] Personalized Image Aesthetics Assessment with Rich Attributes, CVPR, 2022.
>
> [4] Personalized Image Aesthetics Assessment with Attribute-guided Fine-grained Feature Representation, ACM MultiMedia, 2022.

---

> ### Comment · Reviewer_MNoE · 2025-11-27
> **Response to Authors**
>
> I thank the authors for their response.
>
> While the authors argue that the framework's "effectiveness is determined by autonomous editing rather than interactive speed," the paper is targeted at user-facing image editing and optimization; therefore, inference speed remains a critical consideration.
>
> Furthermore, regarding the "effectiveness" emphasized by the authors, I carefully re-examined the quantitative comparison in Section 5.2. The text states: "As shown in Table 1, PhotoAgent achieves the best performance across all evaluation metrics, including CLIP Similarity, LPIPS, BRISQUE, Laion-Reward, and UGC Score." However, I could not find the LPIPS data in Table 1. Moreover, among the five metrics actually listed, the proposed method achieves the best result on only 1 metric and the second-best on 2 metrics. This directly contradicts the claim in the main text that it "achieves the best performance across all evaluation metrics."
>
> Additionally, a runtime comparison with other baseline methods is missing. If the runtime of other methods falls within a reasonable range (e.g., ≤ 30s), I question whether the order-of-magnitude increase in computational cost by PhotoAgent is justified for only marginal performance gains.
>
> These doubts regarding performance significantly heighten my concerns about the novelty and effectiveness of this paper. Consequently, I am tentatively lowering my score to 4.
>
> Please address the inconsistencies regarding Table 1 and Section 5.2.
>
> PS. I noticed that the authors rely solely on $\cite$ for all citations. Please distinguish between textual citations (using $\citet$) and parenthetical citations (using $\citep$) to ensure correct formatting (e.g., 'Author (Year)' vs. '(Author, Year)').

---

> ### Author Response · Authors · 2025-12-02
> **Additional Response to Reviewer MNoE**
>
> Thanks for reviewer's comments.
>
> First, we would like to clarify that our primary focus is establishing a autonomous photo editing system, **not optimizing for speed**. We also provide several methods to accelerate the system with little modification, such as using FP4/FP8 precision, reducing simulation steps, and leveraging TensorRT (Appendix A.1).
>
> Second, our system has **comparable computational time** to other agent-based methods. We acknowledge that fast inference speed is important. While ensuring high-quality output, we provide a detailed analysis of time efficiency and give several acceleration solutions. As shown in Table 4 and Appendix A.1, our method not only maintains superior generation quality but also outperforms some contemporary methods in terms of time cost.
>
> Third, we sincerely thank the reviewer to point out some typos such as LPIPS, which may make inconsistency in the manuscript. We have **thoroughly reviewed** the manuscript to polish the typos and clarity, as well as the citation formatting.

---

### Author Response · Authors · 2025-12-03
**Official Comment by Authors About the Review**

Dear AC，

We sincerely appreciate the time and effort you have dedicated to reviewing our paper. To help you grasp the current status of the review, we would like to provide a summary of the reviewers' comments.

We appreciate that the reviewers consistently acknowledge the effectiveness of our work, specifically highlighting the following strengths:

- The paper is **well-motivated and convincingly** presented (SjC9,Vyiu), **redefines** image editing (MNoE), and **addresses the limitations** of current instruction-based editing models (CZkY).

- The paper **robustly proves the superiority** of the framework (MNoE,CZkY,Vyiu,SjC9). Ablation studies also **clearly validate** the key design choices (MNoE,CZkY,SjC9).

- The UGC-Edit preference dataset and reward model is a **valuable contribution** (MNoE,CZkY,Vyiu). The construction is rigorous, which provides a benchmark and evaluation mechanism that **better captures user-generated content preferences** (MNoE,CZkY,Vyiu).

During the rebuttal phase, we addressed the constructive comments from the reviewers:

- **Computational cost:** We demonstrate that our method achieves comparable time to existing agent-based methods and provide a detailed profiling. We also give practical acceleration strategies to reduce computation cost.

- **Human evaluation:** The user study shows that PhotoAgent is consistently favored, demonstrating strong practical utility and robustness.

- **Dataset diversity and fairness:** We show that the UGC-Edit dataset covers a wide range of scenarios from both English and Chinese domains, ensuring its diversity and fairness.

We once again thank the reviewers and the AC for your time and dedication. We believe that our efforts during the rebuttal, guided by these constructive suggestions, will effectively address the reviewers' concerns, assist the AC in assessing the value of our work, and allow us to enhance the comprehensiveness and rigor of the paper.

Best regards,

Authors of Submission 4224

---

### Meta-Review · Area_Chair_RYT7 · 2026-01-05

**Summary:**

While several reviewers recognized the potential of the proposed aesthetic planning agent, the reviewer consensus remains split due to remaining concerns regarding architectural novelty and missing experimental validations. The AC has carefully reviewed the submission, the rebuttal, and the subsequent discussion. Although the authors provided an initial human user study during the rebuttal phase to address evaluation gaps, it is felt that the concerns regarding the incremental nature of the technical contributions and the computational efficiency of the MCTS framework make it not yet ready for publication. Authors are encouraged to further expand the human-centric evaluations and clarify the algorithmic distinctions from existing modular editing pipelines in future revisions.

**Reviewer Concerns:**

A primary concern raised by multiple reviewers is the limited algorithmic novelty of the framework. Reviewers argued that the system primarily assembles existing components—such as LLM planners, Vision-Language Models (VLM), and off-the-shelf diffusion models—without introducing a fundamental architectural breakthrough in how these models interact or reason. Additionally, while the authors included a preliminary human user study in their response to validate "aesthetic enhancements," some reviewers felt this was insufficient to fully mitigate the perceived reliance on VLM-based metrics. Concerns also persisted regarding the high computational latency inherent in the MCTS-based planning process and whether the framework's performance gains are significantly better than simpler, non-agentic iterative editing baselines when controlled for the same model backends. While the authors provided a rebuttal with new efficiency analyses and the UGC-Edit preference dataset, skepticism remained regarding the system's practical scalability and the distinctiveness of its "exploratory" logic compared to standard task-decomposition methods.

**Reviewer Scores:**

Reviewer CZKY maintained their support for a 6 (Weak Accept), noting the value of the new dataset and the effectiveness of the autonomous decomposition.

Reviewer SjC9, who gave an initial score of 4 (Weak Reject), acknowledged the new efficiency analysis and the preliminary user study; however, their score is likely to stay at a 4 or move at most to a 6 as they remained concerned about the framework's overall novelty.

Reviewer MNOE lowered their score from 6 to a 4 (Weak Reject), explicitly citing high computational latency and data consistency issues in the discussion.

Reviewer Vyiu maintained their positive rating of 6, stating that the rebuttal clarifications and additional user study addressed their concerns.

This leads to likely final scores of 6, 6, 4/6, 4.

---

### Decision · Program_Chairs · 2026-01-26

Reject